# Mouse REC114 is essential for meiotic DNA double-strand break formation and forms a complex with MEI4

Rajeev Kumar[2,*], Cecilia Oliver[1,*], Christine Brun[1,*], Ariadna B Juarez-Martinez[3], Yara Tarabay[1], Jan Kadlec[3], Bernard de Massy[1]

Programmed formation of DNA double-strand breaks (DSBs) initiates the meiotic homologous recombination pathway. This pathway is essential for proper chromosome segregation at the first meiotic division and fertility. Meiotic DSBs are catalyzed by Spo11. Several other proteins are essential for meiotic DSB formation, including three evolutionarily conserved proteins first identified in *Saccharomyces cerevisiae* (Mer2, Mei4, and Rec114). These three *S. cerevisiae* proteins and their mouse orthologs (IHO1, MEI4, and REC114) co-localize on the axes of meiotic chromosomes, and mouse IHO1 and MEI4 are essential for meiotic DSB formation. Here, we show that mouse *Rec114* is required for meiotic DSB formation. Moreover, MEI4 forms a complex with REC114 and IHO1 in mouse spermatocytes, consistent with cytological observations. We then demonstrated in vitro the formation of a stable complex between REC114 C-terminal domain and MEI4 N-terminal domain. We further determine the structure of the REC114 N-terminal domain that revealed similarity with Pleckstrin homology domains. These analyses provide direct insights into the architecture of these essential components of the meiotic DSB machinery.

## Introduction

The conversion from diploid to haploid cells during meiosis requires the expression of a specific and highly differentiated meiotic program in all sexually reproducing eukaryotes. Indeed, meiosis is a specialized cell cycle composed of one replication phase followed directly by two divisions. At the first meiotic division, homologous chromosomes (homologues) are separated through a process called reductional segregation. In most species, reductional segregation requires the establishment of connections between homologues. To achieve this, homologous recombination is induced during meiotic prophase to allow homologues to find each other and to be connected by reciprocal products of recombination (i.e., crossovers) (Hunter, 2015). This homologous recombination pathway is initiated by the formation of DNA double-strand breaks (DSBs) (de Massy, 2013) that are preferentially repaired using the homologous chromatid as a template. Meiotic DSB formation and repair are expected to be tightly regulated because improper DSB repair is a potential threat to genome integrity (Sasaki et al, 2010; Keeney et al, 2014). In *Saccharomyces cerevisiae*, several genes are essential for their formation and at least five of them are evolutionarily conserved. *Spo11*, *Top6bl*, *Iho1*, *Mei4*, and *Rec114* are the mouse orthologs of these five genes (Baudat et al, 2000; Romanienko & Camerini-Otero, 2000; Kumar et al, 2010; Robert et al, 2016a; Stanzione et al, 2016; Tesse et al, 2017) and are specifically expressed in mouse meiotic cells. SPO11 is a homologue of TopoVIA, the catalytic subunit of archea TopoVI, and is covalently bound to the 5′ ends of meiotic DNA breaks. This indicates that meiotic DSBs are formed by a mechanism with similarity to a type II DNA topoisomerase cleavage (Bergerat et al, 1997; Keeney et al, 1997; Neale et al, 2005). SPO11 acts with a second subunit, TOPOVIBL, homologous to archea TopoVIB (Robert et al, 2016a; Vrielynck et al, 2016). TOPOVIBL is quite divergent among eukaryotes, and in some species, such as *S. cerevisiae,* the homologous protein (Rec102) shares only one domain of similarity with TOPOVIBL (Robert et al, 2016a). Based on previous interaction studies between Rec102 and Rec104 (Arora et al, 2004; Jiao et al, 2003), which are both required for meiotic DSB formation, it has been proposed that the *S. cerevisiae* Rec102/Rec104 complex could fulfill the function of TOPOVIBL (Robert et al, 2016b).

The IHO1, MEI4, and REC114 families, which have been shown to be evolutionary conserved (Kumar et al, 2010; Tesse et al, 2017), have been studied in several organisms, including *S. cerevisiae* (Mer2, Mei4, and Rec114), *Schizosaccharomyces pombe* (Rec15, Rec24, and Rec7), *Arabidopsis thaliana* (PRD3, PRD2, and PHS1), *Sordaria macrospora* (Asy2, Mei4 ortholog not identified, and Asy3), *Caenorhabditis elegans* (Mer2 and Mei4 orthologs not identified, and DSB1/2), and *Mus musculus*. Several important properties of

[1]Institut de Génétique Humaine, Centre National de la Recherche Scientifique (CNRS), Université de Montpellier, Montpellier, France  [2]Institut Jean-Pierre Bourgin, Unité Mixte de Recherche 1318 Institut National de la Recherche Agronomique-AgroParisTech, Université Paris-Saclay, Versailles, France  [3]Institut de Biologie Structurale, Université Grenoble Alpes, Commissariat à l'Energie Atomique et aux Energies Alternatives, CNRS, Grenoble, France

Correspondence: bernard.de-massy@igh.cnrs.fr
*Rajeev Kumar, Cecilia Oliver, and Christine Brun contributed equally to this work

these proteins suggest that they act as a complex. Indeed, in *S. cerevisiae* (Li et al, 2006; Maleki et al, 2007) and *M. musculus* (Stanzione et al, 2016), they were shown to co-localize as discrete foci on the axes of meiotic chromosomes which are the structures that develop at the onset of meiotic prophase and allow the anchoring of chromatin loops. Their localization is SPO11-independent, as shown for the three *S. cerevisiae*, for *S. macrospora* Asy2 (Tesse et al, 2017), for the IHO1 and MEI4 *M. musculus* proteins, for *S. pombe* Rec7 (Lorenz et al, 2006), and for *C. elegans* DSB1/2 (Rosu et al, 2013; Stamper et al, 2013). They appear before or at the beginning of meiotic prophase and the number of foci decreases as chromosomes synapse in *S. cerevisiae* (Li et al, 2006; Maleki et al, 2007) and in *M. musculus* (Kumar et al, 2010; Stanzione et al, 2016). In *S. macrospora*, where only Mer2 has been analyzed, its axis localization is also decreased at pachytene upon synapsis (Tesse et al, 2017). In *C. elegans*, foci of the Rec114 orthologs decrease with pachytene progression (Rosu et al, 2013; Stamper et al, 2013). Importantly, in addition to interaction with axis-associated sequences, these proteins are also detected by chromatin immuno-precipitation (ChIP) to interact with DSB sites (Sasanuma et al, 2008; Panizza et al, 2011; Miyoshi et al, 2012; Carballo et al, 2013), which makes sense given their implication in DSB formation. This interaction is weak and transient and is consistent with the tethering of chromatin loops to axes proposed to be established at or before DSB repair (Blat et al, 2002). The determinants of their localization are known to depend on several axis proteins. In *S. cerevisiae*, Mer2, Mei4, and Rec114 are detected particularly at domains enriched in Hop1 and Red1, two interacting meiotic-specific axis proteins (Panizza et al, 2011). The localization of Red1 depends on the meiotic-specific cohesin Rec8 through a direct interaction (Sun et al, 2015).

Molecular organization and activity of the Mer2, Mei4, and Rec114 complex has remained elusive however. Several studies have reported the interactions between these three proteins, suggesting a tripartite complex in *S. pombe* (Steiner et al, 2010; Miyoshi et al, 2012), *S. cerevisiae* (Li et al, 2006; Maleki et al, 2007), and *M. musculus* (Kumar et al, 2010; Stanzione et al, 2016). The current knowledge on their in vivo direct interactions is limited and based only on yeast two-hybrid assays. Mer2 plays a central role and seems to be the protein that allows the recruitment of Mei4 and Rec114 on chromosome axes. This view is based on the observation that the Mer2 orthologs Rec15 in *S. pombe* and IHO1 in *M. musculus*, interact with the axis proteins Rec10 (Lorenz et al, 2006) and HORMAD1 (Stanzione et al, 2016), respectively. In *S. cerevisiae*, Mer2 is necessary for Rec114 and Mei4 recruitment to the axis (Sasanuma et al, 2008) (Panizza et al, 2011). *S. cerevisiae* Mer2 is loaded on chromatin before prophase, during S phase, where it is phosphorylated, a step required for its interaction with Rec114 (Henderson et al, 2006; Murakami & Keeney, 2014). Thus, Mer2 coordinates DNA replication and DSB formation. Analysis of the Mer2 ortholog in *S. macrospora* revealed additional functions in chromosome structure (Tesse et al, 2017). Overall, it is thought that this putative complex (Mer2/Rec114/Mei4) might directly interact with factors involved in the catalytic activity (i.e., at least Spo11/Rec102/Rec104 in *S. cerevisiae*) at DSB sites. Interactions between Rec114 and Rec102 and Rec104 have been detected by yeast two-hybrid assays (Arora et al, 2004; Maleki et al, 2007). Moreover, in *S. pombe,* an additional protein, Mde2 might bridge the Rec15/Rec7/Rec24 and Rec12/Rec6/Rec14

complexes (Miyoshi et al, 2012). However, no specific feature or domain has been identified in Mei4 or Rec114 to understand how they may regulate DSB activity. One could hypothesize that they play a direct role in activating or recruiting the Spo11/TopoVIBL complex for DSB formation. The hypothesis that these proteins might regulate DSB formation through some interactions is also consistent with the findings that Rec114 overexpression inhibits DSB formation in *S. cerevisiae* (Bishop et al, 1999) and that altering Rec114 phosphorylation pattern can up- or down-regulate DSB levels (Carballo et al, 2013). It is possible that Rec114 and Mei4 have distinct roles because Spo11 non-covalent interaction with DSBs is Rec114-dependent but Mei4-independent (Prieler et al, 2005), and Spo11 self-interaction depends on Rec114 but not on Mei4 (Sasanuma et al, 2007). However, in *Zea mays* and *A. thaliana*, the *Rec114* homologue (*Phs1*) seems not to be required for DSB formation (Pawlowski et al, 2004; Ronceret et al, 2009). Here, we performed a functional and molecular analysis to determine whether mouse REC114 is required for meiotic DSB formation, and whether it interacts directly with some of its candidate partners.

## Results

### Rec114-null mutant mice are deficient in meiotic DSB formation

We analyzed mice carrying a null allele of *Rec114*. In the mutated allele (here named Rec114⁻ and registered as Rec114^tm1(KOMP)Wtsi), exon 3 and 4 were deleted, and a lacZ-neomycin cassette with a splice acceptor site was inserted upstream of this deletion. This deletion includes the conserved motifs SSM3, 4, 5, and 6 (Kumar et al, 2010; Tesse et al, 2017) (Figs 1A and S1). We analyzed the cDNA expressed from testes of *Rec114⁻/⁻* mice and showed that the mutant allele is transcribed (Fig S2A). However, because of the presence of a splice acceptor site in the cassette (EnSA, Fig S1), in the cDNA of *Rec114⁻/⁻* mice, exon 2 is fused to DNA sequences from the cassette, themselves fused to exon 5 and 6 but out of frame (Fig S1C). This cDNA from *Rec114⁻/⁻* mice thus encodes for a putative protein containing exon 1 and 2 and lacking all other exons. We conclude that this allele is likely to be a null mutant. We confirmed the absence of detectable REC114 protein in *Rec114⁻/⁻* mice by Western blot analysis of total testes extracts and after REC114 immunoprecipitation (Figs 1B and S2B). Heterozygous (*Rec114⁺/⁻*) and homozygous (*Rec114⁻/⁻*) mutant mice were viable. We also generated from this allele, another mutant allele (named and registered as *Rec114^del*) without the insertion cassette. We performed all subsequent analyses using mice with the *Rec114⁻* allele unless otherwise stated, and confirmed several phenotypes in mice carrying the *Rec114^del* allele.

To monitor the consequences of REC114 absence on gametogenesis, we performed histological analysis of testes and ovaries. Spermatogenesis was altered in *Rec114⁻/⁻* adult male mice, as indicated by the presence of major defects in testis tubule development compared with wild-type (*Rec114⁺/⁺*) mice (Fig 1C). Specifically, in *Rec114⁻/⁻* animals the tubule diameter was smaller and tubules lacked haploid cells (spermatids and spermatozoa). In these tubules, the most advanced cells were spermatocytes, although some were also depleted of spermatocytes. Testis weight was significantly lower in *Rec114⁻/⁻* than wild-type mice (Fig S2C). In

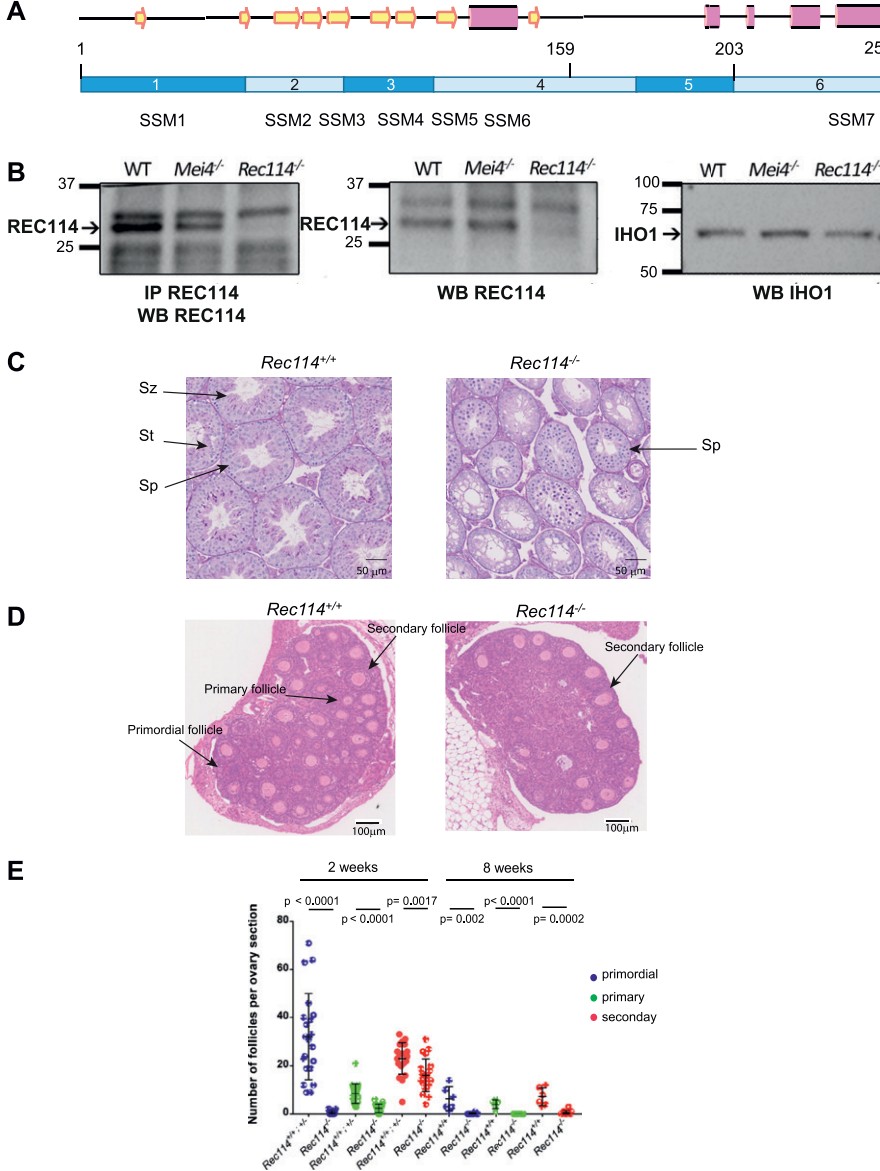

**Figure 1. REC114 is essential for spermatogenesis and oogenesis.**

**(A)** Conserved domains and organization of REC114. The conserved motifs are SSM1 to 7 (Kumar et al, 2010). Secondary structures were predicted with PSIPRED v3.3 (http://bioinf.cs.ucl.ac.uk/psipred/). Pink cylinders, α-helices; yellow arrows, β-sheets. Exons (1 to 6) are shown as dark and light blue rectangles. **(B)** REC114 is not detected in $Rec114^{-/-}$ mice. Western blot (WB) analysis of total testis extracts from WT, $Mei4^{-/-}$, and $Rec114^{-/-}$ as control, prepubertal mice (14 d post-partum, dpp) with anti-REC114 (central panel), with anti-IHO1 (right panel), and anti-REC114 antibodies after immunoprecipitation of REC114 (left panel). **(C)** Spermatogenesis is defective in $Rec114^{-/-}$ mice. Periodic acid-Schiff staining of testis sections from 9-wk-old $Rec114^{+/+}$ and $Rec114^{-/-}$ mice. **(D)** Oogenesis is defective in $Rec114^{-/-}$ mice. Hematoxylin and eosin staining of ovary sections from 2-wk-old $Rec114^{+/+}$ and $Rec114^{-/-}$ mice. **(E)** Quantification of primordial, primary, and secondary follicles in ovaries from 2-wk-old and 8-wk-old $Rec114^{+/+}$, $Rec114^{+/-}$, and $Rec114^{-/-}$ mice. At 2 wk of age, the numbers (mean ± SD) of primordial (blue circles), primary (green circles), and secondary (red circles) follicles were 32.1 ± 17.9, 8.4 ± 4.1, and 23.1 ± 6.5, respectively, for $Rec114^{+/+}$ (n = 3) and $Rec114^{+/-}$ (n = 1) mice ($Rec114^{+/+}$ and $Rec114^{+/-}$ data were pooled, n sections = 21 in total) and 0.6 ± 0.9, 2.3 ± 1.8, and 16.1 ± 6.8, respectively, for $Rec114^{-/-}$ mice (n = 5; n sections = 21). At 8 wk of age, the numbers (mean ± SD) of primordial (blue circles), primary (green circles), and secondary (red circles) follicles were 6.3 ± 5.0, 4 ± 1.8, and 7.2 ± 3.8, respectively, for $Rec114^{+/+}$ mice (n = 1; n sections = 6), and 0.1 ± 0.3, 0 ± 0, and 0.5 ± 1.0, respectively, for $Rec114^{-/-}$ mice (n = 2; n sections = 10). P values were calculated with the Mann–Whitney two-tailed test. Sz, spermatozoa; St, round spermatid; Sp, spermatocyte.

ovaries from $Rec114^{-/-}$ mice, oogenesis was significantly affected, as indicated by the strongly reduced number of primordial follicles at 2 wk postpartum and their near absence at 8 wk (Fig 1D and E). In ovaries from 2 wk old $Rec114^{-/-}$ mice, a significant number of secondary follicles are detected and differentiated from the first wave of follicle growth. These follicles have not been subject to elimination as observed in other DSB-deficient mice (Di Giacomo et al, 2005). Consistent with these gametogenesis defects, $Rec114^{-/-}$ males and females were sterile. Indeed, mating of wild-type C57BL/6 animals with $Rec114^{-/-}$ males and females (n = 3/sex) crossed for 4 months yielded no progeny.

To investigate the nature of the meiotic defect, we monitored by cytological analysis the presence and localization of various markers of recombination and homologous chromosome inter-actions during meiotic prophase. The formation of meiotic DSBs was followed by the detection of γH2AX, the phosphorylated form of H2AX which is enriched in chromatin domains around DSB sites. The DSB repair activity was assessed by the detection of the strand exchange proteins RAD51 and DMC1 and of a subunit (RPA2) of the single-strand DNA-binding protein complex RPA. Chromosome axes and assembly of the synaptonemal complex were monitored by detection of SYCP3 and SYCP1, respectively. Detection of γH2AX revealed that in $Rec114^{-/-}$ mice, meiotic DSBs were absent or strongly reduced in both spermatocytes and oocytes, whereas chromosome axes formed normally, based on SYCP3 detection (Fig 2A). Quantification of the γH2AX signal indicated a 16- and 11-fold reduction in $Rec114^{-/-}$ spermatocytes and oocytes, respectively, compared with wild-type gametocytes (Fig 2B). Consistent with this defect in DSB formation, DSB repair foci were strongly reduced. Specifically, the foci of DMC1 were reduced in $Rec114^{-/-}$ spermatocytes and oocytes compared with wild-type cells (Figs 2C and D, and S3). Similarly, RPA2 and RAD51 foci were strongly reduced or

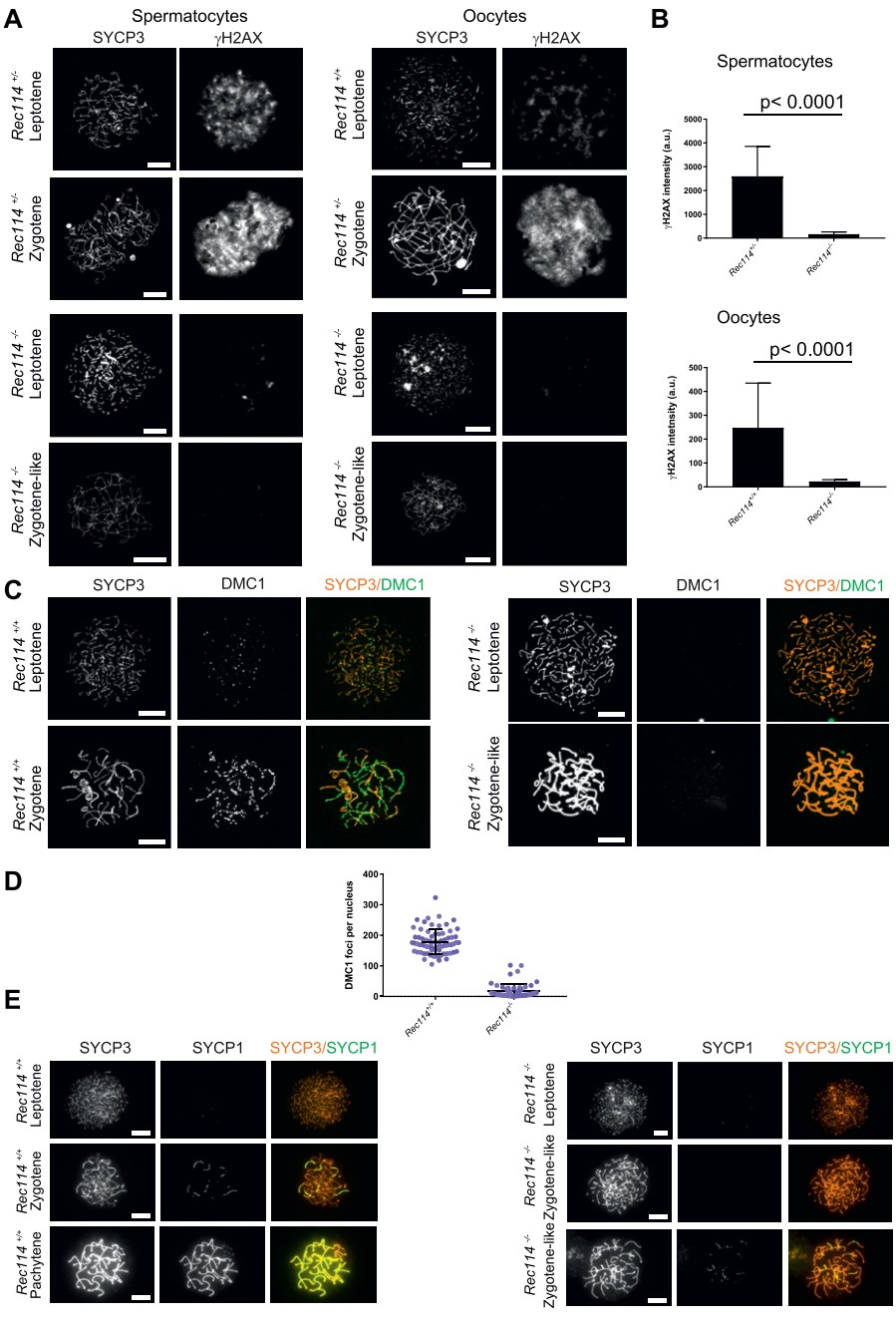

**Figure 2. _Rec114⁻/⁻_ mice show defects in DSB formation and homologous synapsis.**
**(A)** Immunostaining of γH2AX and SYCP3 in spermatocytes from 13 dpp _Rec114⁺/⁻_ and _Rec114⁻/⁻_ males, and from E15 (15 d of embryonic development) _Rec114⁺/⁺_, _Rec114⁺/⁻_, and _Rec114⁻/⁻_ oocytes. In _Rec114⁻/⁻_ spermatocytes and oocytes, no pachynema could be observed and spermatocytes or oocytes with partially synapsed chromosomes were defined as zygotene-like. Scale bar, 10 μm. **(B)** Quantification of the total γH2AX signal per nucleus (mean ± SD; a.u, arbitrary units) on spreads from leptotene spermatocytes (13 dpp) and from leptotene oocytes (E15): 2,597 ± 1,261 and 165 ± 95 in _Rec114⁺/⁻_ and _Rec114⁻/⁻_ males, respectively (n = 53 and 50); 248 ± 187 and 23 ± 7 in _Rec114⁺/⁺_ and _Rec114⁻/⁻_ females, respectively (n = 48 and 47). P values were calculated with the Mann–Whitney two-tailed test. **(C)** Immunostaining of DMC1 and SYCP3 in spermatocytes from 15 dpp _Rec114⁺/⁺_ and _Rec114⁻/⁻_ males. Scale bar, 10 μm. **(D)** Quantification of DMC1 foci (mean ± SD) in leptotene and zygotene spermatocytes from _Rec114⁺/⁺_ and _Rec114⁻/⁻_ mice (178.6 ± 39.9 and 17.2 ± 23.0 in _Rec114⁺/⁺_ and _Rec114⁻/⁻_ males, respectively; n = 71 and 59). P < 0.0001 (Mann–Whitney two-tailed test). **(E)** Immunostaining of SYCP1 and SYCP3 in spermatocytes from 15 dpp _Rec114⁺/⁺_ and _Rec114⁻/⁻_ mice. Scale bar, 10 μm.

undetectable in _Rec114⁻/⁻_ compared with wild-type gametocytes (Figs S4A and S5A). Although foci were detected with the anti-DMC1 antibody in _Rec114⁻/⁻_ oocytes, we interpret those as nonspecific because monitoring RPA2 foci showed a strongly reduced level of DSB repair foci consistent with the strong reduction of γH2AX (Fig S4B). The detection of SPO11 by immunoprecipitation in _Rec114⁻/⁻_ mice showed that the meiotic DSB formation deficiency in these mice is not due to absence of SPO11 (Fig S5B). Meiotic DSB formation and repair promotes interactions between homologues that are stabilized by the loading of SYCP1, a component of the synaptonemal complex (Fraune et al, 2012). Analysis of SYCP1 localization during meiosis showed major defects in both male and female

_Rec114⁻/⁻_ meiocytes. The presence of short SYCP1 stretches suggested progression into zygonema; however, these stretches never elongated to form a full-length synaptonemal complex between homologues, indicating failure of homologous synapsis formation (Figs 2E and S6). Altogether, the phenotypes of _Rec114⁻/⁻_ mice are highly similar to those of the previously characterized _Spo11⁻/⁻_ (Baudat et al, 2000; Romanienko & Camerini-Otero, 2000), _Mei1⁻/⁻_ (Libby et al, 2002, 2003), _Mei4⁻/⁻_ (Kumar et al, 2010), _Iho1⁻/⁻_ (Stanzione et al, 2016), and _Top6bl⁻/⁻_ (Robert et al, 2016a) mice where the formation of DSBs, DSB repair foci, and of homologous synapses are strongly affected. Histological and cytological analyses of the _Rec114^{del/del}_ mutant mice showed similar

phenotypes, indicating that the cassette present in the *Rec114*⁻ allele does not cause the observed meiotic defects (Fig S7).

### In vivo REC114 interacts with MEI4 and these proteins display a mutually dependent localization

REC114, MEI4, and IHO1 co-localize on the axis of meiotic chromosomes, and IHO1 is needed for MEI4 loading (Stanzione et al, 2016). First, we tested whether IHO1 loading required REC114 and MEI4. This was clearly not the case because IHO1 localization was similar in wild type and in *Rec114*⁻/⁻ and *Mei4*⁻/⁻ spermatocytes (Figs 3A and S8A). This observation is consistent with a role for IHO1 in REC114 and MEI4 recruitment.

We then tested whether MEI4 and REC114 regulated each other's localization. MEI4 forms 200–300 foci on meiotic chromosome axes at leptonema. Then, the focus number progressively decreases as cells progress into zygonema, and MEI4 becomes undetectable at

pachynema. This decrease of MEI4 foci during meiotic progression is directly correlated with synapsis formation (MEI4 foci are specifically depleted from synapsed axes) and with DSB repair (MEI4 foci are excluded from DMC1 foci) (Kumar et al, 2010). At leptonema, the number of axis-associated MEI4 foci was reduced 3- to 4-fold in *Rec114*⁻/⁻ spermatocytes and oocytes (Figs 3B and C, S8B, and S9A), and their intensity was significantly decreased (by 1.75-fold in spermatocytes and by 1.9-fold in oocytes) compared with wild-type controls (Fig S9B). The MEI4 signal detected in *Rec114*⁻/⁻ gametocytes was higher than the nonspecific background signal observed in *Mei4*⁻/⁻ spermatocytes (Fig 3C). This suggests that REC114 contributes to, but it is not essential for MEI4 focus formation on meiotic chromosome axis.

REC114 foci co-localize with MEI4 and, like MEI4 foci, their number is highest at leptonema and then progressively decreases upon synapsis (Stanzione et al, 2016). We thus tested whether REC114 foci required MEI4 for axis localization. At leptonema, few

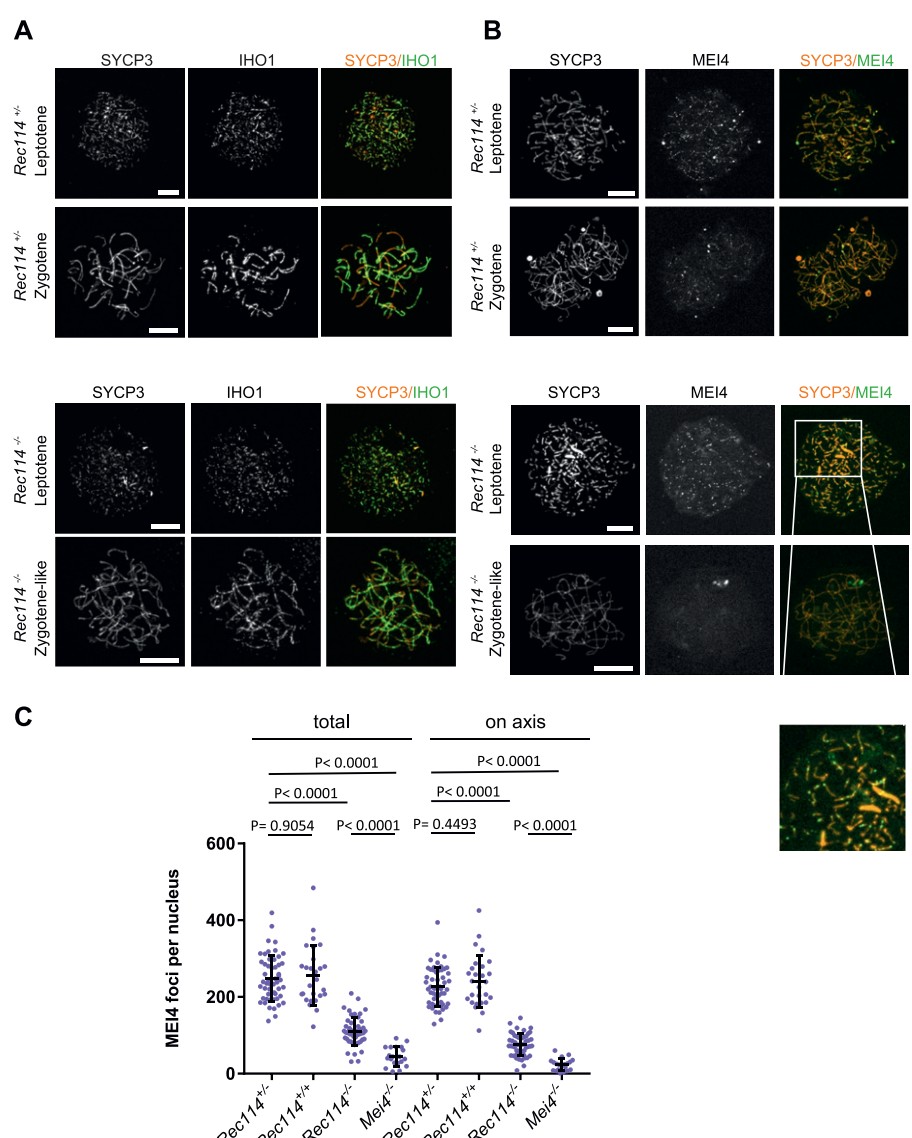

**Figure 3. REC114 is required for robust MEI4 foci localization.**
**(A)** Immunostaining of IHO1 and SYCP3 in early prophase spermatocytes from 13 dpp *Rec114*⁺/⁻ and *Rec114*⁻/⁻ males. Scale bar, 10 μm. **(B)** Immunostaining of MEI4 and SYCP3 in early prophase spermatocytes from 13 dpp *Rec114*⁺/⁻ and *Rec114*⁻/⁻ males. Scale bar, 10 μm. **(C)** Quantification of MEI4 foci in leptotene spermatocytes from 13 dpp *Rec114*⁺/⁻, *Rec114*⁺/⁺, *Rec114*⁻/⁻, and *Mei4*⁻/⁻ males. The numbers (mean ± SD) of total foci and of foci on chromosome axes (co-localized with SYCP3) were: 248 ± 59 and 226 ± 51 for *Rec114*⁺/⁻ (n = 53 nuclei), 256 ± 78 and 241 ± 68 for *Rec114*⁺/⁺ (n = 27), 109 ± 37 and 76 ± 29 for *Rec114*⁻/⁻ (n = 50), 45 ± 25, and 23 ± 16 for *Mei4*⁻/⁻ (n = 20), respectively. *P* values were calculated with the Mann–Whitney two-tailed test.

axis-associated REC114 foci above the background signal could be detected in $Mei4^{-/-}$ spermatocytes, where their number was reduced by more than 10-fold compared with wild-type cells (Fig 4A and B). However, this low level of REC114 foci in $Mei4^{-/-}$ was still significantly higher compared with the number in $Rec114^{-/-}$ gametocytes (Fig 4B). REC114 foci were not reduced in $Spo11^{-/-}$ mice (Fig 4B), as previously reported for MEI4 foci (Kumar et al, 2010). This indicates that these proteins are loaded on the chromosome axis independently of SPO11 activity. Overall, MEI4 and REC114 are reciprocally required for their localization.

MEI4 and REC114 co-localization, their mutual dependency for robust localization, and their interaction in yeast two-hybrid assays (Kumar et al, 2010) strongly suggested that these two proteins interact directly or indirectly in vivo. Indeed, we could detect REC114 after immunoprecipitation of MEI4 in extracts from wild-type and

$Spo11^{-/-}$ mice (Fig 4C). These assays were performed using protein extracts from 14 dpp mice where cellular composition in the testes are similar in wild type and mutants deficient for DSB formation. In this assay, we observed that the amount of MEI4 recovered after immunoprecipitation with the anti-MEI4 antibody is reduced in extracts from $Rec114^{-/-}$. This observation may indicate a destabilization of MEI4 in the absence of REC114. In principle, this could also be analyzed by detecting proteins from a total extract, but MEI4 is not detectable in such extracts in our conditions. As REC114 was detected in total protein extracts, we could observe that the REC114 level is not altered in the absence of MEI4 (Figs 1 and S9C). Interestingly, IHO1 also was immunoprecipitated by the anti-MEI4 antibody suggesting that MEI4 interacts with both IHO1 and REC114 (Fig 4C). These three proteins could be a part of the same complex, or form two independent complexes. Although MEI4 and IHO1 did

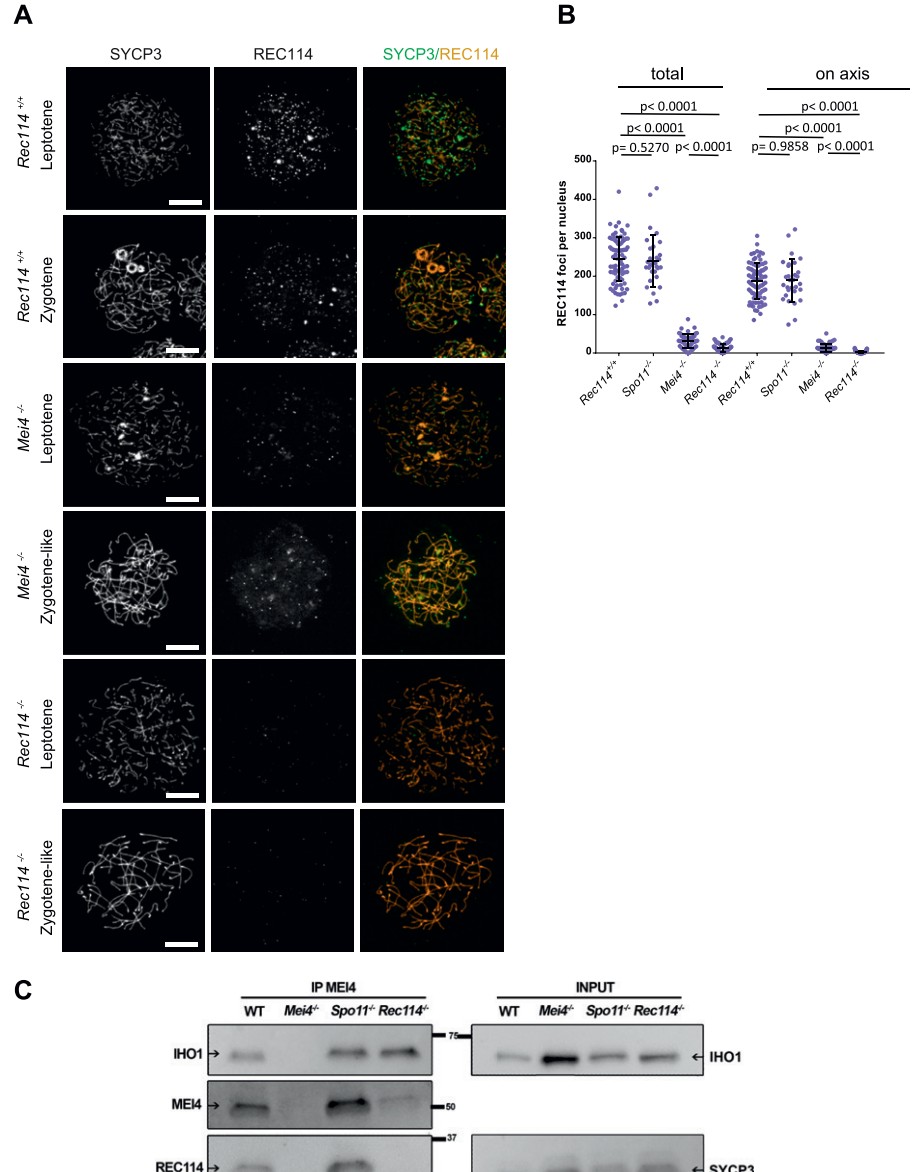

**Figure 4. MEI4, REC114 and IHO1 form a complex.**
**(A)** Immunostaining of REC114 and SYCP3 in early prophase spermatocytes from $Rec114^{+/+}$ (14 dpp), $Mei4^{-/-}$ (12 dpp), and $Rec114^{-/-}$ (14 dpp) males. Scale bar, 10 μm. **(B)** Quantification of REC114 foci in leptotene spermatocytes from 13 dpp $Rec114^{+/+}$, $Mei4^{-/-}$, $Spo11^{-/-}$, and $Rec114^{-/-}$ males. The numbers (mean ± SD) of total foci and of foci on chromosome axis (co-localized with SYCP3) were: 245 ± 58 and 187 ± 47 for $Rec114^{+/+}$ (n = 83 nuclei), 241 ± 68 and 189 ± 55 for $Spo11^{-/-}$ (n = 30), 13 ± 11 and 3 ± 3 for $Rec114^{-/-}$ (n = 69), and 32 ± 17 and 14 ± 11 for $Mei4^{-/-}$ (n = 52). P values were calculated with the Mann–Whitney two-tailed test. **(C)** Co-immunoprecipitation of REC114 and IHO1 with MEI4. Total testis extracts from 14 dpp $Rec114^{+/+}$ (WT), $Mei4^{-/-}$, $Spo11^{-/-}$, and $Rec114^{-/-}$ mice were immunoprecipitated with an anti-MEI4 antibody. Input extracts were probed with anti-IHO1 and anti-SYCP3 antibodies. MEI4 is not detected in input extracts. Immunoprecipitated fractions were probed with anti-IHO1, anti-MEI4, and anti-REC114 antibodies.

not interact in a yeast two-hybrid assay (Stanzione et al, 2016), the detection of IHO1 after immunoprecipitation of MEI4 in *Rec114*$^{-/-}$ extracts (Fig 4C) suggests a direct or indirect interaction between IHO1 and MEI4 in mouse spermatocytes. We note that after immunoprecipitation of MEI4, the level of IHO1 is not reduced in *Rec114*$^{-/-}$ compared with the wild type, whereas the level of MEI4 is. Formally, this could be explained by an excess of MEI4 relative to the IHO1 protein available for interaction with MEI4. Alternatively, changes in protein complex structure in the absence of REC114 could lead to modification of interactions and their recovery in the conditions used for immunoprecipitation.

### REC114 and MEI4 form a stable complex

These in vivo assays suggested that REC114 and MEI4 directly interact. To test whether these two proteins interact and form a stable complex in vitro, we produced recombinant full-length REC114 in bacteria (Fig 5). However, we could not produce recombinant full-length MEI4 or its N-terminal or C-terminal domains alone. Conversely, when co-expressed with REC114, the N-terminal fragment (1–127) of MEI4 was soluble and could be co-purified with REC114 on Strep-Tactin resin (Fig 5A, lane 4), providing the first biochemical evidence of a direct interaction between REC114 and MEI4. To identify the REC114 region that interacts with MEI4, we produced the N-terminal domain and a C-terminal fragment (residues 203–254) of REC114 and found that the REC114 C-terminal region (but not the N-terminal domain) was sufficient for binding to MEI4 (Figs 5A, lanes 5, 6 and S10). Finally, we could purify the MEI4 (1–127) and REC114 (203-254) complex and show that the two proteins co-eluted as a single peak from the Superdex 200 gel filtration column (Fig 5B and C).

### Rec114 contains a pleckstrin homology (PH) domain

To gain insights into the structure of mouse REC114, we produced the full-length protein in bacteria. Then, using limited trypsin proteolysis, we identified a stable fragment (residues 15–159) that was suitable for structural analysis. We determined the crystal structure of this REC114 N-terminal region at a resolution of 2.5 Å by SAD using the selenomethionine (SeMet)-substituted protein. The final model, refined to an $R_{free}$ of 30% and $R$-factor of 25% included residues 15–150 (Table 1).

Unexpectedly, the structure revealed that REC114 (15-150) forms a PH domain, with two perpendicular antiparallel $\beta$-sheets followed by a C-terminal helix (Fig 6). Several residues are disordered in loops between the $\beta$ strands. In the SeMet protein dataset that we solved at 2.7 Å, the crystallographic asymmetric unit contained two REC114 molecules, but the position of $\beta$2 that packs against $\beta$1 in one of the molecules was shifted by three residues. A Protein Data Bank search using the PDBeFold server at EBI revealed that REC114 (15-150) was highly similar to the other PH domains and that the N-terminal domain of the CARM1 arginine methyltransferase (PDB code 2OQB) was the closest homologue (Fig S11).

Mapping the conserved residues to the protein surface revealed that both $\beta$-sheets contained exposed conserved residues that could be involved in protein interactions with REC114 partners (Fig S12). In the crystal, the PH domain formed extensive crystallographic contacts with a symmetry-related molecule. Indeed, this interface, judged as significant using the PDBePisa server, buried a surface of 764 Å$^2$ and included several salt bridges and hydrogen bonds formed by the well-conserved Arg98 of $\beta$6, and Glu130 and Gln137 of $\alpha$1 (Fig S13A). To test whether REC114 dimerized in solution, we analyzed the PH domain by size-exclusion chromatography–multiple angle laser light scattering (SEC-MALLS) that allows measuring the molecular weight. Although the monomer molecular mass of the fragment was 16 kD (32 kD for a dimer), the MALLS data indicated a molecular weight of 24.7 kD for the sample at the concentration of 10 mg/ml. When injected at lower concentrations, the protein eluted later and the molecular weight diminished (21 kD at 5 mg/ml) (Fig S13B). These results could be explained by a concentration-dependent dimerization of the PH domain with a

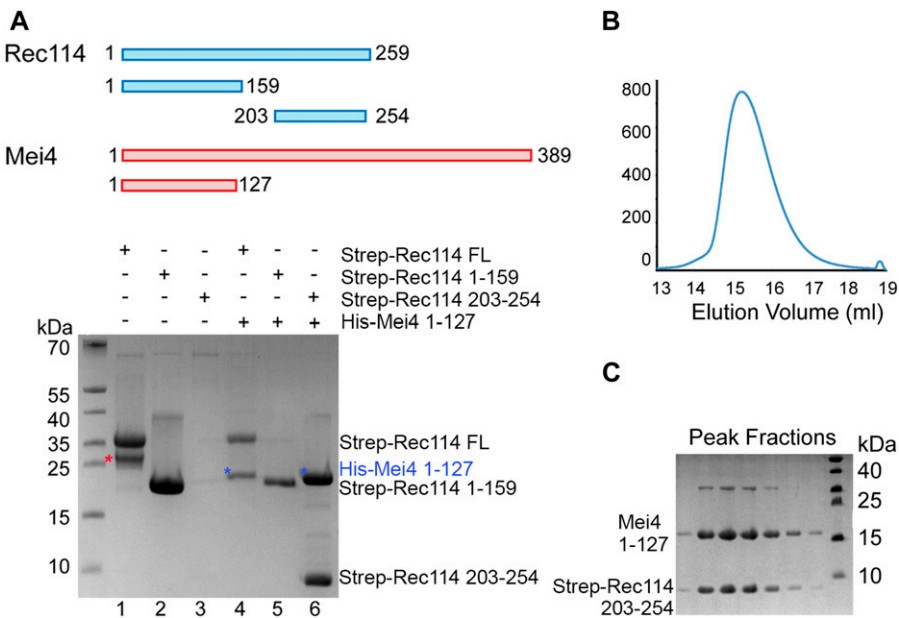

**Figure 5. REC114 forms a stable complex with MEI4.**
A Strep-tag pull-down experiments with the REC114 and MEI4 constructs are shown in the upper panel. Full length and fragments of Strep-REC1114 were purified alone or after co-expression with MEI4 (1-127). Strep-REC114 (203-254) was insoluble on its own (lane 3), but when co-expressed with MEI4 (1-127) became soluble and was sufficient for interaction with MEI4 (lane 6). MEI4 (1-127, blue star) was pulled down by full-length (FL) REC114 and REC114 (203-254), but not by REC114 (1-159) (lanes 4–6). Proteins were detected by Coomassie blue staining. **(A)** Western blot confirmation of His-Mei4 with anti-His antibody is shown in Fig S10. Red star indicates a degradation fragment of FL REC114. **(B)** Strep-tagged REC114 (203-254) was co-expressed with His-MEI4 (1-127) and purified first using Strep-Tactin resin. The His-tag of MEI4 was removed by TEV protease cleavage followed by a passage through a Ni$^{2+}$ column. The complex was then purified by Superdex 200 size-exclusion chromatography. The gel filtration elution profile is shown. **(C)** SDS–PAGE analysis of the peak fractions shown in (B). Proteins were detected by Coomassie blue staining.

**Table 1.** Data collection and refinement statistics for REC114 (15–159).

|  | REC114 native | REC114 SeMet |
|---|---|---|
| Data collection | | |
| Space group | $P6_122$ | $P4_22_12$ |
| Cell dimensions | | |
| $a, b, c$ (Å) | 107.5, 107.5, 82.8 | 88.5, 88.5, 101.5 |
| $\alpha, \beta, \gamma$ (°) | 90, 90, 120 | 90, 90, 90 |
| Resolution (Å) | 93–2.5 (2.57–2.5)[a] | 66.7–2.7 (2.8–2.7) |
| $R_{merge}$ | 5.7 (212.1) | 12.2 (108.8) |
| $I/\sigma I$ | 24.2 (1.11) | 11.1 (1.68) |
| $CC_{1/2}$ | 1 (0.548) | 0.997 (0.725) |
| Completeness (%) | 99.6 (97.5) | 100 (99.9) |
| Redundancy | 11.2 (11.1) | 7.1 (7.1) |
| Refinement | | |
| Resolution (Å) | 46.6–2.5 | |
| No. reflections | 10,215 | |
| $R_{work}/R_{free}$ | 25 (30) | |
| $B$-factors | 77.3 | |
| R.m.s. deviations | | |
| Bond lengths (Å) | 0.014 | |
| Bond angles (°) | 1.641 | |

[a]Values in parentheses are for the highest resolution shell.

fast exchange rate between monomers and dimers that co-purify together during SEC. The validation of the monomer interface and the significance of this putative dimerization will be interesting to evaluate.

# Discussion

Previous studies in yeast have shown that the putative complex involving *S. cerevisiae* Rec114, Mei4, and Mer2 is essential for meiotic DSB formation. Their transient localization at the DSB sites (observed by ChIP) suggests that, in yeast, this complex may play a direct role in promoting DSB activity (Sasanuma et al, 2007, 2008; Panizza et al, 2011; Carballo et al, 2013). Several studies have shown the evolutionary conservation of these three partners. In mammals, MEI4 and IHO1 (the Mei4 and Mer2 orthologs, respectively) are required for meiotic DSB formation (Kumar et al, 2010; Stanzione et al, 2016). Here, we show that REC114 function in the formation of meiotic DSBs is conserved in the mouse. Moreover, we provide the first direct evidence of the interaction between REC114 and MEI4 and identified a potential interaction domain in REC114 that includes previously identified conserved motifs.

## Properties of REC114

Our study revealed that REC114 N-terminus is a PH domain that is composed of two sets of perpendicular anti-parallel $\beta$-sheets followed by an $\alpha$ helix. This domain is present in a large family of proteins with diverse biological functions and is mostly involved in targeting proteins to a specific site and/or in protein interactions. A subset of these proteins interacts with phosphoinositide phosphates (Lietzke et al, 2000; Lemmon, 2003). Several conserved positively charged residues in the two $\beta$ sheets, $\beta1$ and $\beta2$, important for the interaction are not present in REC114. However, subsequent studies revealed interactions between the PH domain and a variety of different partners, in some cases by binding to phosphotyrosine-containing proteins or to polyproline (Scheffzek & Welti, 2012). Therefore, the REC114 PH domain could be a platform for several interactions, some of which could involve phosphorylated serine or threonine residues because it has been shown that ATR/ATM signaling through phosphorylation of downstream proteins regulate meiotic DSB activity (Joyce et al, 2011; Lange et al, 2011; Zhang et al, 2011; Carballo et al, 2013; Cooper et al, 2014).

In terms of conservation of the REC114 primary sequence, it is remarkable that most of the previously described conserved motifs (SSM1 to 6) are structural elements ($\beta$ sheets and $\alpha$ helices) within this PH domain and are readily identified in many eukaryotes (Kumar et al, 2010; Tesse et al, 2017). At the REC114 C-terminus, the SSM7 motif overlaps with a predicted $\alpha$ helical structure and is less well conserved. Moreover, its presence remains to be established in several species (Tesse et al, 2017). In this study, we demonstrated that this C-terminal domain directly interacts with MEI4, suggesting that this SSM7 region is evolutionarily conserved. The N-terminal domain of MEI4 that interacts with REC114 has a predicted $\alpha$ helical structure and includes two conserved motifs (Kumar et al, 2010).

## Interaction of REC114 with the chromosome axis

A previous study showed that IHO1 is required for MEI4 and REC114 focus formation on axis and that it directly interacts with REC114 by two-hybrid assay (Stanzione et al, 2016). As IHO1 interacts with HORMAD1, IHO1 could act as a platform to recruit REC114 and MEI4. Such a mechanism would be similar the one identified in *S. cerevisiae* for the recruitment of Rec114 and Mei4 by Mer2 (Henderson et al, 2006; Panizza et al, 2011). In agreement with this hypothesis, IHO1 association with the chromosome axis is not altered in the absence of MEI4 or REC114, similar to that observed in *S. cerevisiae* (Panizza et al, 2011). Therefore, IHO1 could recruit REC114 by direct interaction, and this should allow MEI4 recruitment. Alternatively, we suggest a mechanism where REC114/MEI4 would be recruited as a complex to the axis as we observed a mutual dependency between these two proteins for their axis localization: the formation of REC114 axis-associated foci is strongly reduced in the absence of MEI4 and reciprocally. The residual REC114 foci detected in the absence of MEI4 do not appear to be able to promote DSB formation as DSB repair foci are abolished in *Mei4$^{-/-}$* mice similar to *Spo11$^{-/-}$* mice, and thus, suggesting an active role for the REC114/MEI4 complex. MEI4 may also be able to interact (directly or indirectly) with IHO1 or with axis proteins independently from REC114 at least in an *Rec114$^{-/-}$* genetic background as weak MEI4 axis-associated foci were observed in *Rec114$^{-/-}$* spermatocytes and oocytes and because IHO1 protein was detected upon immunoprecipitation of MEI4 in *Rec114$^{-/-}$* spermatocyte extracts. The details of these interactions and their dynamics during early meiotic prophase remain to be analyzed.

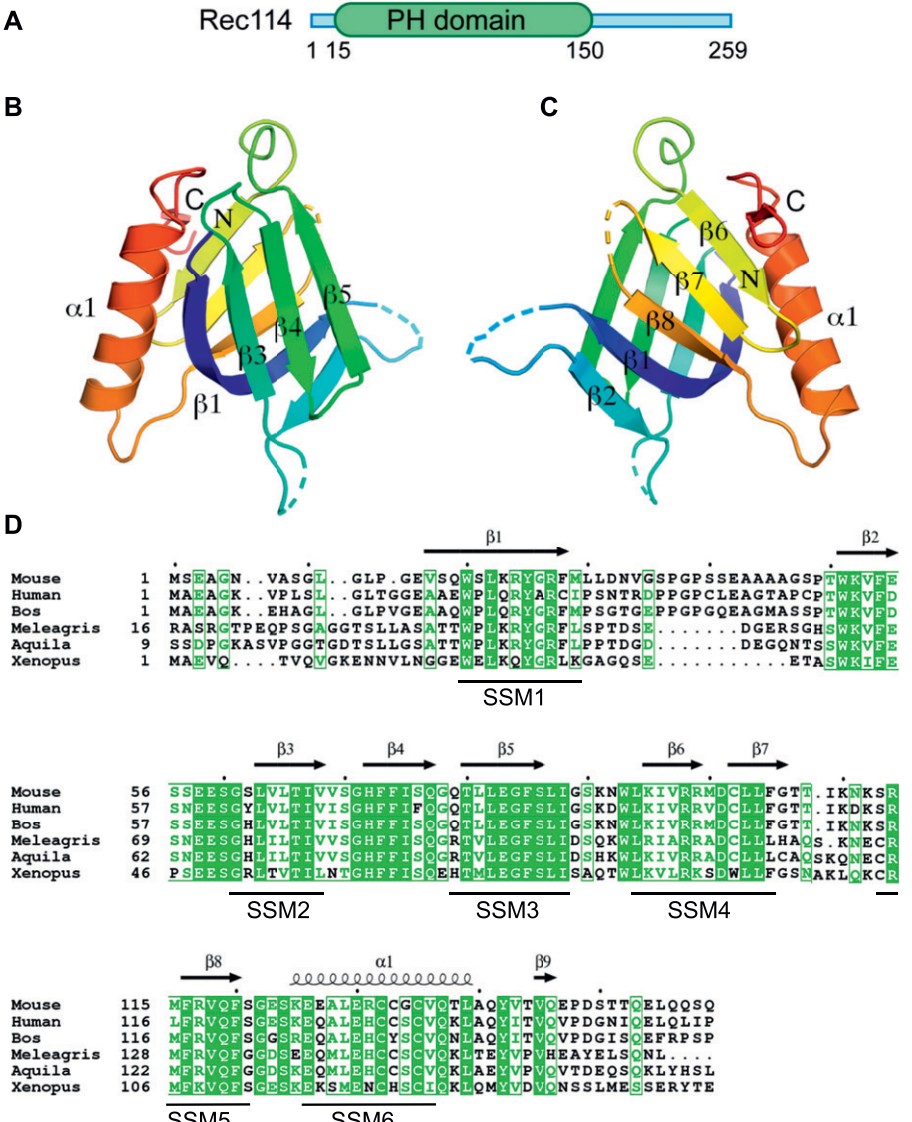

**Figure 6. Crystal structure of the REC114 PH domain.**
**(A)** Schematic representation of the PH domain structure in mouse REC114, based on this work. **(B)** Ribbon diagram of the REC114 PH domain. The polypeptide chain is colored from the N-terminus (blue) to the C-terminus (red). Missing residues in the loops are shown by dashed lines. **(C)** The same ribbon diagram as in (A), but rotated 180° around the vertical axis. **(D)** Sequence alignment of REC114 proteins. Residues that are 100% conserved are in solid green boxes. The secondary structures of REC114 are shown above the sequences. The previously identified conserved motifs, SSM1 to SSM6 (Kumar et al, 2010; Tesse et al, 2017), are shown below the sequence. Mouse, *Mus musculus*: NP_082874.1; Human, *Homo sapiens*: NP_001035826.1; Bos, *Bos mutus*: XP_005907161.1; Meleagris, *Meleagris gallopavo*: XP_019474806.1; Aquila, *Aquila chrisaetos canadensis*: XP_011595470.1; Xenopus, *Xenopus laevis*: OCT89407.1.

Overall, the IHO1/MEI4/REC114 complex is expected to be the main component for the control of SPO11/TOPOVIBL catalytic activity. It may be essential for turning on and off the catalytic activity. In *S. cerevisiae*, it has been proposed that the local control of meiotic DSB formation is constrained by the chromatin loop organization and involves Tel1 (ATM) (Garcia et al, 2015) and possibly also the Mer2/Mei4/Rec114 (IHO1/MEI4/REC114) complex. Indeed, *S. cerevisiae* Rec114 shows that Tel1/Mec1-dependent phosphorylation is associated with down-regulation of DSB activity (Carballo et al, 2013). The IHO1/MEI4/REC114 complex could be a limiting factor for DSB formation. In agreement, we noted that the number of cytologically detectable foci is of the same order (about 200) as the number of DSB events measured by the detection of DSB repair proteins. The shutting off of DSB formation that correlates with synapsis between homologues (Thacker et al, 2014) could be the direct consequence of the removal of the Hop1 (or HORMAD1 in mice) axis protein, resulting in the displacement of the Mer2/Mei4/

Rec114 (IHO1/MEI4/REC114 in mice) complex from the axis. Additional studies on the protein–protein interactions and posttranslational modifications will help to understand these important steps for the regulation of meiotic DSB formation.

# Materials and Methods

### Mouse strains

The non-conditional *Rec114* mutant allele is referenced as 2410076I21Rik^tm1(KOMP)Wtsi and named *Rec114*^−/− in this study. The *Rec114*^del^ allele, in which the inserted lacZ-Neo cassette was deleted, was obtained by expression of Flp in mice carrying *Rec114*^−/−. These mice are in the C57BL/6 background. The *Mei4*^−/−, *Spo11*^−/−, and *Spo11*^YF/YF^ strains were previously described (Baudat et al, 2000; Kumar et al, 2010; Carofiglio et al, 2013). All animal

experiments were carried out according to the CNRS guidelines and approved by the ethics committee on live animals (project CE-LR-0812 and 1295).

## cDNA analysis

Total RNA was extracted from two testes of young (14 dpp) wild type or Rec114$^{-/-}$ mice using the GeneJet RNA Purification Kit (Thermo Fisher Scientific) according to the manufacturer's instructions. One $\mu$g of total RNA was treated with RQ1 RNAse-free DNase (Promega) 30 min at 37°C to degrade genomic DNA. 1 $\mu$g of total RNA was used for cDNA synthesis, using random primers and the Transcriptor First Strand cDNA Synthesis Kit (Roche) according to the manufacturer's instructions. Each PCR was performed using 0.05 $\mu$l of cDNA and KAPA Taq PCR kit (Sigma-Aldrich). Oligonucleotides sequences are provided in the Table S1. PCR cycling conditions were 3 min at 95°C, 35 cycles for 16U22/232L22 and 695U22/806L25 or 40 cycles for 16U21/644L22 and 16U21/688L24 with 15 s at 95°C, annealing for 15 s at 53°C, 51°C, 64°C, and 59°C for 16U22/232L22, 695U22/806L25, 16U21/644L22, and 16U21/688L24, respectively, and extension for 30 s at 72°C, followed by 5 min at 72°C at the end of cycles. Amplified products were separated on 2% agarose gels. PCR products obtained from the oligonucleotide combinations 16U21/644L22 or 16U21/688L24 were run in a gel and the most abundant product purified and sequenced.

## Antibodies

Chicken anti-REC114 antibodies were generated against mouse REC114 and affinity-purified. The anti-SYCP3, anti-MEI4, and anti-IHO1 antibodies were previously described (Baudat & de Massy, 2007; Kumar et al, 2010; Stanzione et al, 2016). Other antibodies used in this study were against $\gamma$H2AX (Millipore 05-636), DMC1 (SC-22768; Santa Cruz), SYCP1 (ab15090; Abcam), RPA32 (gift from R. Knippers), and RAD51 (gift from W. Baarends). For immunofluorescence experiments, the following secondary antibodies were used: Cy3 AffiniPure donkey anti-guinea pig IgG (H+L) (Jackson ImmunoResearch), donkey anti-rabbit IgG (H+L) Alexa Fluor 488 (Thermo Fisher Scientific), and donkey anti-mouse IgG (H+L) Alexa Fluor 647 (Thermo Fisher Scientific).

## Histology and cytology

Testes and ovaries were fixed in Bouin's solution (Sigma-Aldrich) at room temperature overnight and for 5 h, respectively. After dehydration and embedding in paraffin, 3-$\mu$m sections were prepared and stained with periodic acid-Schiff for testes and with hematoxylin and eosin for ovaries. Image processing and analysis were carried out with the NDP.view2 software (Hamamatsu).

Spermatocyte and oocyte chromosome spreads were prepared by the dry-down method (Peters et al, 1997).

## Image analysis

$\gamma$H2AX was quantified using Cell profiler 2.2.0. The total pixel intensity per nucleus was quantified. The intensity of MEI4 foci was the mean pixel value within a focus. Axis-associated MEI4 foci were determined by co-labelling with SYCP3.

## Protein analysis

Whole testis protein extracts were prepared as described in Stanzione et al (2016).

For REC114 immunoprecipitation, 1.5 mg of extract (from 14 dpp mice) was diluted in IP buffer (20 mM Tris-HCl; 150 mM NaCl; 0.05% NP-40; 0.1% Tween-20; 10% glycerol; protease inhibitors) and incubated with 2 $\mu$g of affinity-purified chicken anti-REC114 antibody at 4°C overnight. Then, 50 $\mu$l of agarose-immobilized anti-chicken IgY Fc (goat) (GGFC-130D, Icllab) was added at 4°C for 1 h. Beads were washed five times with washing buffer (20 mM Tris-HCl, pH 7.5, 0.05% NP-40, 0.1% Tween-20, 10% glycerol, 150 mM NaCl). Immunoprecipitated material was eluted and incubated with 2× Laemmli loading buffer (with 20 mM DTT) at 95°C for 5 min.

For MEI4 immunoprecipitation, 3 $\mu$g of guinea pig anti-MEI4 antibody (Stanzione et al, 2016) was cross-linked to 1.5 mg of Dynabeads Protein A (Invitrogen) with disuccinimidyl suberate using the Crosslink Magnetic IP/Co-IP Kit (Pierce; Thermo Fisher Scientific). 3.6 mg of testis protein extract (from 14 dpp mice) was incubated with the cross-linked antibody at 4°C overnight. Beads were washed five times with washing buffer (20 mM Tris–HCl, pH 7.5, 0.05% NP-40, 0.1% Tween-20, 10% glycerol, 150 mM NaCl). Immunoprecipitated material was eluted by incubating the beads with the elution buffer (pH 2) for 5 min, and neutralized with the neutralization buffer (pH 8.5) (both buffers provided with the kit). The eluates were incubated with Laemmli loading buffer (1× final) at RT for 10 min, and divided in three aliquots, adding 10 mM DTT to one of them (for REC114 detection), followed by incubation at 95°C for 5 min.

SPO11 immunoprecipitations on protein extracts from adult mice were performed as described (Pan & Keeney, 2009). Rabbit polyclonal anti-SPO11 was used for detection (Carofiglio et al, 2013).

## Western blot analysis

Immunoprecipitates and inputs were separated on 10% Mini-PROTEAN TGX precast gels (Bio-Rad) and then transferred onto nitrocellulose membranes with the Trans-Blot Turbo Transfer System (Bio-Rad). The following primary antibodies were used: rabbit anti-MEI4 (1/500; Kumar et al, 2010), chicken anti-REC114 (1/1,000), rabbit anti-IHO1 (1/2,000; Stanzione et al, 2016), and guinea pig anti-SYCP3 (1/3,000; Kumar et al, 2010). The following HRP-conjugated secondary antibodies were used: anti-rabbit (1:5,000; Cell Signaling Technology), True-Blot anti-rabbit (1/1,000; Jackson ImmunoResearch), donkey anti-chicken IgY (1/3,000; Jackson ImmunoResearch), and donkey anti-guinea pig (1/10,000; Jackson ImmunoResearch).

## Protein expression, purification, and crystallization

Mouse REC114 (15-159) fused to His-tag was expressed in *E. coli* BL21-Gold (DE3) (Agilent) from the pProEXHTb expression vector (Invitrogen). The protein was first purified by affinity chromatography using Ni$^{2+}$ resin. After His-tag cleavage with the TEV protease,

the protein was further purified through a second $Ni^{2+}$ column and by size-exclusion chromatography. Pure protein was concentrated to 10 mg·ml$^{-1}$ in buffer (20 mM Tris, pH 7.0, 200 mM NaCl, and 5 mM mercaptoethanol). The best-diffracting crystals grew within 1 wk at 20°C in a solution containing 0.25 M ammonium sulfate, 0.1 M MES, pH 6.5, and 28% PEG 5000 MME. For data collection at 100 K, crystals were snap-frozen in liquid nitrogen with a solution containing mother liquor and 25% (vol/vol) glycerol. SeMet-substituted REC114 was produced in *E. coli* BL21-Gold (DE3) and a defined medium containing 50 mg·l$^{-1}$ of SeMet. SeMet REC114 was purified and crystallized as for the native protein.

### Data collection and structure determination

Crystals of REC114 (15-159) belong to the space group $P6_122$ with the unit cell dimensions $a$, $b$ = 107.5 Å and $c$ = 82.8 Å. The asymmetric unit contains one molecule and has a solvent content of 71%. A complete native dataset was collected to a resolution of 2.5 Å by the autonomous ESRF beamline MASSIF-1 (Bowler et al, 2015). The SeMet REC114 crystallized in the same conditions in the space group $P4_22_12$ and contained two molecules per asymmetric unit. A complete SeMet dataset was collected to a resolution of 2.7 Å at the peak wavelength of the Se K-edge on the ID23-1 beamline at the ESRF. Data were processed using XDS (Kabsch, 2010). The structure was solved using SeMet SAD data. Selenium sites were identified, refined, and used for phasing in AUTOSHARP (Bricogne et al, 2003). The model was partially built with BUCCANEER (Cowtan, 2006), completed manually in COOT (Emsley et al, 2010) and refined with REFMAC5 (Murshudov et al, 1997). The model was used for molecular replacement to determine the structure using the native dataset and PHASER (McCoy et al, 2007). The native structure was finalized in COOT and refined with REFMAC5 to a final $R$-factor of 25% and $R_{free}$ of 30% (Table 1) with all residues in the allowed regions (96% in favored regions) of the Ramachandran plot, as analyzed using MOLPROBITY (Chen et al, 2010).

### Strep-tag pull-down assays

MEI4 (1-127) was cloned in the pProEXHTb expression vector to produce a His-tag fusion protein. REC114 and its deletion mutants were cloned in the pRSFDuet-1 vector as Strep-tag fusion proteins. REC114 variants alone or co-expressed with MEI4 were purified using a Strep-Tactin XT resin (IBA). The resin was extensively washed with a buffer containing 20 mM Tris, pH 7.0, 200 mM NaCl, and 5 mM mercaptoethanol, and bound proteins were eluted with the same buffer containing 50 mM biotin and analyzed by 15% SDS–PAGE. The minimal REC114-MEI4 complex was then purified using the Strep-Tactin XT resin. The His-tag of MEI4 was removed with the TEV protease and a passage through a $Ni^{2+}$ column. The complex was then purified by size-exclusion chromatography.

## Supplementary Information

## Acknowledgments

We thank Yukiko Imai for a lot of advice on immunoprecipitation assays and Frédéric Baudat for help and advice on histology. We thank all the laboratory members for insight and discussions. We thank Amélie Sarazin for image analysis. We thank Scott Keeney for providing anti-SPO11 antibody. We thank Attila Toth for comments on the manuscript, and providing anti-IHO1 and anti-MEI4 antibodies. We thank Morgane Auboiron for help in immunocytochemistry. B de Massy thanks Akira Shinohara and Osaka University for providing support during the preparation of the manuscript. We thank the following BioCampus Montpellier facilities: the Réseau des Animaleries de Montpellier for animal care, the Réseau d'Histologie Expérimentale de Montpellier for histology, the Montpellier Resources Imagerie for microscopy, and the TAAM/CNRS facility. AB Juarez-Martinez was supported by the Labex GRAL (Grenoble Alliance for Integrated Structural Cell Biology) (ANR-10-LABX-49-01). This work used the platforms of the Grenoble Instruct-ERIC Center (ISBG: UMS 3518 CNRS-CEA-UGA-EMBL) with support from FRISBI (French Infrastructure for Integrated Structural Biology) (ANR-10-INSB-05-02) and GRAL (ANR-10-LABX-49-01) within the Grenoble Partnership for Structural Biology (PSB). We thank Caroline Mas and Marc Jamin for assistance with MALLS and Luca Signor for mass spectrometry analysis. We thank the staff of the ESRF-EMBL (European Synchrotron Radiation Facility-European Molecular Biology Laboratory) Joint Structural Biology Group, particularly Matthew Bowler, for access to and help with the ESRF beamlines. We thank the EMBL high-throughput crystallization facility (HTX). B de Massy was funded by grants from the Centre National pour la Recherche Scientifique and the European Research Council Executive Agency under the European Community's Seventh Framework Programme (FP7/2007–2013 Grant Agreement no. [322788]). C Oliver was funded in part by a postdoctoral fellowship from LabexEpigenMed, program « Investissements d'avenir », ANR-10-LABX-12-01. B de Massy was the recipient of the Prize Coups d'Élan for French Research from the Fondation Bettencourt-Schueller.

### Author Contributions

R Kumar initiated the project, analyzed the mice, and performed the cytological analysis.
C Oliver analyzed the mice and protein interactions.
C Brun analyzed the mice, performed the cytological analysis, and quantified the data.
AB Juarez-Martinez performed the in vitro assays.
Y Tarabay prepared the samples and antibodies.
J Kadlec designed and performed the in vitro assays and structural analysis.
B de Massy supervised the project, analyzed the data, prepared the figures, and wrote the manuscript.

### Conflict of Interest Statement

The authors declare that they have no conflict of interest.

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
