## [Reviewer comments · Life Science Alliance]

Life Science Alliance

Mouse REC114 is essential for meiotic DNA double-strand break and forms a complex with MEI4

Bernard de Massy, Rajeev Kumar, Cecilia Oliver, Christine Brun, Ariadna Juarez-Martinez, Yara Tarabay, and Jan Kadlec
DOI: 10.26508/lsa.201800259

Corresponding author(s): Bernard de Massy, IGH, Centre National de la Recherche Scientifique, Univ Montpellier

Review Timeline:	Submission Date:	2018-11-29
	Editorial Decision:	2018-11-30
	Revision Received:	2018-12-04
	Accepted:	2018-12-04

Scientific Editor: Andrea Leibfried

Transaction Report:

Please note that the manuscript was previously reviewed at another journal and the reports were taken into account in the decision-making process at Life Science Alliance. Since the original reviews are not subject to Life Science Alliance's transparent review process policy, the reports and author response cannot be published.

November 30, 2018

RE: Life Science Alliance Manuscript #LSA-2018-00259-TR

Dr. Bernard de Massy
Institute of Human Genetics
UMR 9002 CNRS UM 141 rue de la cardonille
Montpellier 34396

Dear Dr. de Massy,

Thank you for submitting your revised manuscript entitled "Mouse REC114 is essential for meiotic DNA double-strand break and forms a complex with MEI4" to Life Science Alliance.

Your work had been previously reviewed at another journal, and those reports were transferred to us with your permission. The reviewers who evaluated your work thought that your findings are of high quality, corroborate and extend our knowledge on REC114 (homologs), and that they will be of interest to others in the field. Based on those reports already at hand, we invited you to submit a revised version of your work to LSA, including controls and clarifications requested by the reviewers as well as more data to support the absence of DSBs and better support for the absence of Rec114 in the knock-out condition. You have provided such a revised version as well as a full point-by-point response to the concerns raised at the other journal. We appreciate your response and the changes made to the manuscript, and we would be happy to publish your paper in Life Science Alliance pending final revisions necessary to meet our formatting guidelines:

- please add the number of replicates performed (in either method section or in figure legends)
- please note that the S figures will be displayed in-line in the HTML version of your paper. We would therefore appreciate a single page displaying all figure panels (Figures S1-S4 run over several pages currently)

A. FINAL FILES:

-- High-resolution figure, supplementary figure and video files uploaded as individual files: See our detailed guidelines for preparing your production-ready images, <http://life-science-alliance.org/authorguide>

B. MANUSCRIPT ORGANIZATION AND FORMATTING:

Full guidelines are available on our Instructions for Authors page, <http://life-science-alliance.org/authorguide>

Sincerely,

December 4, 2018

RE: Life Science Alliance Manuscript #LSA-2018-00259-TRR

Dr. Bernard de Massy
IGH, Centre National de la Recherche Scientifique, Univ Montpellier
UMR 9002 CNRS UM 141 rue de la cardonille
Montpellier 34396

Dear Dr. de Massy,

Thank you for submitting your Research Article entitled "Mouse REC114 is essential for meiotic DNA double-strand break and forms a complex with MEI4". It is a pleasure to let you know that your manuscript is now accepted for publication in Life Science Alliance. Congratulations on this interesting work.

DISTRIBUTION OF MATERIALS:

Again, congratulations on a very nice paper. I hope you found the review process to be constructive and are pleased with how the manuscript was handled editorially. We look forward to future exciting submissions from your lab.

Sincerely,
Andrea Leibfried, PhD
Executive Editor
Life Science Alliance
Meyerhofstr. 1
69117 Heidelberg, Germany
t +49 6221 8891 502